# Estradiol and Estrone Have Different Biological Functions to Induce NF-κB-Driven Inflammation, EMT and Stemness in ER+ Cancer Cells

**DOI:** 10.3390/ijms24021221

**Published:** 2023-01-07

**Authors:** Ana Belén Diaz-Ruano, Nuria Martinez-Alarcon, Macarena Perán, Karim Benabdellah, María de los Ángeles Garcia-Martinez, Ovidiu Preda, César Ramirez-Tortosa, Andrea Gonzalez-Hernandez, Juan Antonio Marchal, Manuel Picon-Ruiz

**Affiliations:** 1Biopathology and Regenerative Medicine Institute (IBIMER), Centre for Biomedical Research, University of Granada, 18100 Granada, Spain; 2Excellence Research Unit “Modeling Nature” (MNat), University of Granada, 18016 Granada, Spain; 3BioFab i3D Laboratory—Biofabrication and 3D (bio) Printing Laboratory, 18100 Granada, Spain; 4Department of Health Sciences, University of Jaén, 23071 Jaén, Spain; 5GENYO Centre for Genomics and Oncological Research, Genomic Medicine Department, Pfizer/University of Granada/Andalusian Regional Government, Health Sciences Technology Park, Av. de la Illustration 114, 18016 Granada, Spain; 6University Hospital San Cecilio, Av. del Conocimiento, s/n, 18016 Granada, Spain; 7Instituto de Investigación Biosanitaria de Granada (ibs.GRANADA), 18012 Granada, Spain; 8Department of Human Anatomy and Embryology, Faculty of Medicine, University of Granada, 18016 Granada, Spain

**Keywords:** ER+ cancer, estradiol, estrone, inflammation, NF-κB, HSD17B

## Abstract

In general, the risk of being diagnosed with cancer increases with age; however, the development of estrogen-receptor-positive (ER+) cancer types in women are more closely related to menopausal status than age. In fact, the general risk factors for cancer development, such as obesity-induced inflammation, show differences in their association with ER+ cancer risk in pre- and postmenopausal women. Here, we tested the role of the principal estrogens in the bloodstream before and after menopause, estradiol (E2) and estrone (E1), respectively, on inflammation, epithelial-to-mesenchymal transition (EMT) and cancer stem cell enrichment in the human ER+ cervical cancer cell line HeLa. Our results demonstrate that E1, contrary to E2, is pro-inflammatory, increases embryonic stem-transcription factors (ES-TFs) expression and induces EMT in ER+ HeLa cells. Moreover, we observed that high intratumoural expression levels of 17β-Hydroxysteroid dehydrogenase (HSD17B) isoforms involved in E1 synthesis is a poor prognosis factor, while overexpression of E2-synthetizing HSD17B isoforms is associated with a better outcome, for patients diagnosed with ER+ ovarian and uterine corpus carcinomas. This work demonstrates that E1 and E2 have different biological functions in ER+ gynaecologic cancers. These results open a new line of research in the study of ER+ cancer subtypes, highlighting the potential key oncogenic role of E1 and HSD17B E1-synthesizing enzymes in the development and progression of these diseases.

## 1. Introduction

Estrogens play a key role in the regulation of multiple physiological processes such as sexual development, stress response, tissue differentiation, regulation of energy metabolism, etc. [1,2]. Throughout a woman’s life, the concentration and type of estrogen in circulation substantially vary. However, the most important change occurs during menopause, when the ovaries stop producing estradiol (E2) and estrogens are mainly synthesized in peripheral tissues, such as the adipose tissue, in the form of estrone (E1) [1,3,4]. This change in the source of estrogens causes E1 to replace E2 as the principal estrogen in the bloodstream [3,4]. 

Several diseases and disorders in women are closely associated with the estrogenic milieu. Among them, the most important are those related to cancer. Estrogen-dependent cancer subtypes are the most commonly diagnosed cancers among women worldwide, accounting for at least 30% of all diagnosed cancer cases, and the first cause of cancer death in this population, with up to 20% of total cancer-related deaths [5]. It is important to note that the main risk factors for the development of estrogen-dependent cancer subtypes, including those of the breast, ovary, cervix or endometrium [6,7,8], are directly related to estrogen synthesis. These risk factors include obesity, age, menopausal status or prolonged use of oral contraceptives, among others [9,10,11]. Interestingly, the risk for sporadic (non-hereditary) estrogen-dependent cancer development increases markedly when E1 becomes the major serum estrogen, which occurs after menopause, and particularly in women with high serum E1 levels, which corresponds to the obese postmenopausal women cohort [12]. 

In the last decades, the incidence of these diseases has increased markedly, which may be explained in part by the increase in obesity worldwide [13]. Interestingly, several epidemiological studies indicate that the relation between obesity and estrogen-dependent cancer risk depends on menopausal status. For example, in breast cancer, obesity is consistently associated with a higher risk of postmenopausal breast cancer, but it is not considered a risk factor in premenopausal women [14]. Furthermore, even some recent epidemiological studies associate obesity with lower breast cancer risk in the premenopausal state [15,16]. Moreover, this association seems to be restricted to ER-positive (ER+) subtypes, suggesting a link between obesity, estrogens and ER+ cancer risk [17]. In fact, these observations could be explained by the role that the adipose tissue plays in two critical biological processes: inflammation and estrogen biosynthesis.

Regarding the role of obesity in inflammation, it is well known that the obese adipose tissue creates an inflammatory environment that promotes the development and progression of different cancers, including breast, ovarian and endometrial cancer [15,16,17,18,19]. In this microenvironment, the proinflammatory M1 macrophage subtype induces expression of inflammatory factors such as tumour necrosis factor alpha (TNF-α), interleukin (L)-6, IL-8, IL-1β, chemokine (C-C motif) ligand 2 (CCL2) and vascular endothelial growth factor (VEGF), which recruit immune cells and support vasculogenesis. Moreover, free fatty acids (FFAs), released in obese individuals, stimulate toll-like receptor 4 (TLR4) and activate the nuclear factor kappa B (NF-κB) pathway, which are mainly responsible for obese adipose tissue inflammation. NF-κB upregulates the secretion of several of the previously listed cytokines, creating a positive feed-forward loop to sustain chronic inflammation and cancer progression [20,21,22]. 

The relationship between obesity and estrogen synthesis is particularly important after menopause. Before menopause, estrogens are mainly synthesized by the ovaries in the form of E2. However, after menopause, the principal estrogen during the reproductive phase, E2, is reduced and E1 becomes the main serum estrogen, produced mostly by peripheral conversion of androstenedione by aromatase, which is abundant in the adipose tissue and whose levels increase with obesity [17,22]. Thus, estrogen levels in postmenopausal obese women, the cohort with the highest risk to develop estrogen-dependent cancers, are particularly high for E1 [23,24].

It has been assumed for decades that E2 and E1, the main estrogens before and after menopause, respectively, had similar biological functions but differed in their estrogen receptor alpha (ERα) binding affinity, i.e., greater for E2 than E1, considering E2 as the active form of estrogens. Thus, almost all published works in this field have been limited to studying the effect of E2 on ER+ cancer diseases. Regarding the role of E2 on obesity-mediated NF-κB regulation, many studies have shown that E2-bound ERα opposes NF-κB signaling by several mechanisms and in different cell types [24]. However, we have recently demonstrated that E2 and E1 have different roles in NF-κB pathway regulation and ER+ breast cancer progression [25,26]. 

In this study, we investigated whether E2 and E1 play different roles in the progression of other estrogen-dependent cancer types in addition to breast cancer, such as cervix, uterus and ovary cancers. Moreover, we evaluated the clinical value for the patient outcome of several HSD17B isoforms, a family of enzymes involved in the conversion of E2 into E1 and vice versa. This work demonstrates that E1 and E2 have different biological functions in ER+ human cervical cancer cells. E1, contrary to E2, is pro-inflammatory, increases embryonic stem-transcription factors (ES-TFs) expression and induces EMT in ER+ HeLa cells. Moreover, high intratumoural expression levels of HSD17B isoforms involved in E1 synthesis is a poor prognosis factor, while overexpression of E2-synthetizing HSD17B isoforms is associated with a better outcome for patients diagnosed with ER+ ovarian and uterine corpus carcinomas. These results open a new line of research in the study of ER+ cancer subtypes, highlighting the potential key oncogenic role of E1 and HSD17B E1-synthesizing enzymes in the development and progression of these diseases. 

## 2. Results

### 2.1. Estrogen Receptor Is Expressed in HeLa Cells

HeLa cells have been proven to express both estrogen and progesterone receptors [27]. In any case, the gene expression of the two isomers, ERα (*ESR1*) and ERβ (*ESR2*), in HeLa cells was assayed using qPCR. In addition, the expression of the ERα transcript (66 KDa) was confirmed using Western Blot and immunofluorescence. As shown in Figure 1A, both ER isomers were detected in HeLa cells at mRNA levels, with a higher expression for ERα than ERβ; however, both were expressed at low levels. Moreover, ERα was also detected at protein level in this cell line (Figure 1B).

### 2.2. TNF-α Exposure Activates the NF-κB Signalling Pathway in HeLa Cells

It is well known that the NF-κB pathway is activated in several cell types by TNF-α exposure, which is released by the obese adipose tissue. For this experiment, HeLa cells were grown in the presence of 10 ng/mL TNF-α for 45 min and NF-κB pathway activation was determined using NF-κB transcription factors translocation to the nucleus with immunofluorescence. TNF-α treatment induced nuclear translocation of all the five proteins of the NF-κB family: RelA, RelB, c-Rel, p50 and p52 (Figure 2A–E). These results demonstrate that TNF-α treatment activates the NF-κB pathway in HeLa cells. 

### 2.3. The Activation of the NF-κB Signalling Pathway Induces the EMT Process in HeLa Cells

To determine the role of the NF-κB pathway in EMT, we performed a wound-healing assay. Figure 3A,B shows how TNFα-treated HeLa cells were able to completely close the wound after 3 days, while only 56% of wound healing closure was observed for the control non-treated cells. It is important to note that faster wound healing closure may not be due to an activation of the EMT but due to increased cell replication. To further demonstrate that these results were due to an activation of the EMT, we studied the effect of TNFα treatment on cell growth. As shown in Figure 3C, cell growth rates for both the control and treated HeLa cells were not statistically different, demonstrating that the better healing ability of treated cells was because of the TNFα effect on EMT.

### 2.4. Estrone, but Not Estradiol, Increases TNF-α-Induced Inflammation in HeLa Cells

Anti-inflammatory activities have been related to E2 in several diseases [25]; however, the role of E1 in a pro-inflammatory environment associated with obesity is unknown. To test how E1 and E2 are involved in NF-κB-mediated inflammation, HeLa cells were exposed to TNF-α for 4 h with the purpose to recapitulate the pro-inflammatory environment associated with obesity. Cells were also exposed to a combination of TNF-α and E1 or E2, and the expression of several pro-inflammatory cytokines was assayed using qPCR. As expected, based on the results shown in Section 2.2, TNF-α significantly increased the expression levels of all these cytokines compared to the control (EtOH), inducing an inflammatory response in HeLa cells (Figure 4A). Interestingly, E1 significantly increased TNF-α-upregulation of IL-6, while E2 did not (Figure 4A). Regarding CCL2 and IL-8 expression, E1 seemed to increase and E2 decrease their expression, particularly for IL-8. However, no statistically significant differences were found due to experimental variations between replicates (Figure 4A). 

### 2.5. TNF-α and Estrone, but Not Estradiol, Increase Stemness Properties of HeLa Cells

Self-renewal ability is a characteristic of both normal and cancer stem cells and is mainly driven by embryonic stem cell transcription factors (ES-TFs). Firstly, we tested using qPCR analysis the expression of the ES-TF genes c-MYC, KLF4, OCT, NANOG and SOX2 in HeLa cells after long exposure (3 weeks) to E1 or E2. However, we only obtained statistically significant results for the expression of c-MYC, SOX2 and KLF4 markers. As shown in Figure 4B, c-MYC, KLF4 and SOX2 were significantly upregulated in HeLa cells exposed to E1, while E2 did not change or even seem to slightly decrease the levels of these ES-TFs.

In addition, to determine the effect of TNF-α treatment alone or in combination with E1 or E2 in stem cell enrichment in HeLa cells, we studied changes in ALDH activity using flow cytometry. Cells exposed to TNF-α for a long term were lightly enriched in cells ALDH^+^ (49.43 ± 7.9%) with respect to control cells (38.41 ± 7.1%) (Figure 4C). On the other hand, E1 seemed to maintain the TNF-α induced enrichment (50.23 ± 9%) and E2 tended to reduce this ALDH enrichment (33.53 ± 9.3%) (Figure 4C). 

### 2.6. Estrone, but Not Estradiol, Induces Epithelial-to-Mesenchymal Transition in HeLa Cells

Epithelial-to-mesenchymal transition (EMT) is a process required for the mobility of epithelial cells, which is closely related to metastasis in cancer. To determine the role of E1 and E2 in EMT progression, we tested using qPCR the expression of different EMT main transcription factors. As shown in Figure 4D, long exposure to E1 by itself upregulated the expression of the EMT transcription factors SNAIL and TWIST1 in HeLa cells, while E2 did not. However, other markers such as E-Cad, N-Cad, Vimentin or SLUG did not show any difference with respect to the control.

### 2.7. Generation of HeLa pERα Cells

Since HeLa cells express low levels of ERs, cells were transfected with a cDNA pERα plasmid to generate a stable cell line overexpressing ERα (HeLa pERα) (Figure 5A,B). To determine that ERα overexpression successfully increased the responsiveness of HeLa cells to estrogens, we compared the upregulation of GREB1, a well-known direct target gene of ERα, after treatment with E2 in HeLa WT and pERα cells. As expected, GREB1 levels were increased in HeLa cells transfected with pERα with respect to HeLa WT cells when stimulated with the ERα ligand E2 (Figure 5C).

### 2.8. TNF-α and Estrone, but Not Estradiol, Increases ES-TF Expression in HeLa pERα Cells

The expression of some ES-TFs was analysed to study the self-renewal ability of HeLa pERα cells after long exposure to treatment with TNF-α alone or in combination with E1 or E2. Figure 5B shows that c-MYC and KLF4 expression were significantly increased in HeLa pERα cells exposed to TNF-α + E1 with respect to the control (EtOH) and TNF-α treated cells. In contrast, E2 exposure induced the downregulation of c-MYC expression both when compared to the control and to the cells treated with TNF-α alone (Figure 5C,D). KLF4 levels were also lower for TNF-α + E2 than the levels achieved with TNF-α + E1, although, KLF4 levels did not change compared to the control, there were significantly higher levels for TNF-α + E1 compared to TNF-α treatment alone (Figure 5B). On the contrary, the case of short-term exposure (one week) to 10 nM, E1 or E2 did not seem to show significant differences in other markers, such as SOX2, OCT and NANOG. 

In addition to qPCR analysis, ALDH cytometry was also performed. Unfortunately, the difference between the groups treated for a period of one week and the control group did not show any difference.

### 2.9. Estrone Induces Epithelial-to-Mesechymal Transition in HeLa pERα Cells

We tested the effect on EMT upregulation of E1 and E2 exposure after NF-κB activation using TNF-α treatment in HeLa pERα cells. Expression of the EMT transcription factors SLUG, TWIST1 and SNAIL was first analysed using qPCR. Figure 5C shows that TNF-α in combination with E1 upregulated these three transcription factors, reaching levels higher than observed for TNF-α treatment alone. Interestingly, SNAIL expression was significantly downregulated in TNF-α + E2 cells with respect to the cells treated only with TNF-α; however, expression levels of SLUG and TWIST1 were slightly higher for TNF-α + E2 cells than for TNF-α treated cells. In addition, the expression of the EMT markers Vimentin and N-Cadherin was also analysed. The profile expression for these two EMT markers was the same as the results obtained for SLUG and TWIST1 (Figure 5D–F). E1 in combination with TNF-α significantly increased the expression levels of both markers, whereas E2 showed a modest increase of SLUG and TWIST when compared to the results obtained by TNF-α treatment alone (Figure 5D).

### 2.10. HSD17B Enzymes Involved in Estrone Synthesis Have an Oncogenic Role in Hormone-Dependent Cancer Diseases

We have previously demonstrated that high HSD17B14 expression, an enzyme which catalyses the conversion of E2 into E1, is associated with tumour progression and worse patient outcomes in ER+ breast cancer [25]. Here, we analysed public clinical data to determine if E1-synthesizing HSD17B enzymes are relevant for other estrogen-dependent cancer diseases. We focused our studies on the analysis of HSD17B2, HSD17B4, HSD17B6, HSD17B10 and HSD17B14, that are known to have a preferential oxidative activity related to an increase in E1 synthesis [28,29]. 

The results obtained from these analyses demonstrate that E1 HSD17B-synthesizing enzymes play a key role in hormone-dependent cancerous diseases. In this sense, we found hypomethylation of the HSD17B2 promoter and overexpression of this enzyme in uterine corpus endometrial carcinoma (UCEC) compared to normal tissue (Figure 6A,B). In addition, we also found an overexpression of the HSD17B10 enzyme in endometrial carcinomas of the uterine body compared to normal tissue (Figure 6C).

The KM Plotter^TM^ human ovarian cancer primary database was analysed to assess the expression of different HSD17B enzymes involved in E1 synthesis to predict the outcome of ovarian cancer patients for three clinical parameters: progression-free survival (PFS), post-progression survival (PPS), and overall survival (OS). High intratumoural mRNA expression levels of the oxidative HSD17B enzymes HSD17B2, HSD17B4, HSD17B6 and HSD17B14, involved in E1 synthesis, correlate with lower PFS (HR = 0.77 (0.68–0.88), *p* = 0.00011; HR = 1.25 (1.08–1.43), *p* = 0.0021; HR = 1.44 (1.25–1.66), *p* = 5.5 × 10^−7^; HR = 1.3 (1.12–1.5), *p* = 0.00051; respectively) (Figure 6D–G). In addition, increased expression of the E1 synthetic enzymes HSD17B2, HSD17B4 and HSD17B6 in ovarian tumours is inversely associated with the probability of PPS (HR = 1.21 (1–1.45), *p* = 0.045; HR = 1.29 (1.07–1.55), *p* = 0.0073; respectively) (Figure 6D–F). Moreover, elevated intratumoural E1 synthesis by HSD17B2, HSD17B4, HSD17B6 and HSD17B14 overexpression in primary ovarian tumours correlate with lower OS (Figure 6D–G). 

### 2.11. HSD17B Enzymes Involved in Estradiol Synthesis Have a Protection Role in Hormone-Dependent Cancer Diseases

We also analysed the KM Plotter^TM^ primary human ovarian cancer database for the correlation between the expression levels of HSD17B enzymes involved in E2 synthesis and patient outcome. Contrary to the results observed for E1-synthetizing enzymes, high expression levels of HSD17B enzymes that increase E2 levels are mainly associated with better patient outcomes in ovarian cancer. 

Increased intratumoural expression levels of the reducing enzyme HSD17B5, which promotes the conversion of E1 into E2, showed a positive association with the probability of PFS, therefore representing a good prognostic factor (HR = 0.81 (0.71–0.93), *p* = 0.0027) (Figure 7A). Higher expression levels of the reducing enzymes HSD17B1 and HSD17B12, which increase E2 levels, are associated with a lower probability of PPS (HR = 0.76 (0.64–0.9), *p* = 0.0012; and HR = 0.79 (0.66–0.96), *p* = 0.016, respectively) as shown in Figure 7B,C. In addition, high intratumoural mRNA levels of the enzyme HSD17B7, which increases E2 levels, are associated with increased OS in ovarian cancer patients (HR = 0.83 (0.72–0.94), *p* = 0.0052) (Figure 7D). 

In addition, the analysis of “The Cancer Genome Atlas” (TCGA) database shows a lower expression of the E2-synthetizing enzymes HSD17B12 and HSD17B5 (also called AKR1C3) in tumour tissue samples compared to normal tissues for UCEC (Figure 7E,F). These data support the results obtained from in vitro studies that demonstrate a differential role of both estrogens, E1 and E2, in inflammation and ER+ cancer development and progression. 

### 2.12. Low Expression of HSD17B Enzymes Involved in Estradiol Synthesis Combined with High Expression of HSD17B Enzymes Involved in Estrone Synthesis Correlate with a Worse Patient Outcome in ER+ Ovarian Cancer Diseases

Finally, we analysed the KM Plotter^TM^ primary human ovarian cancer database for the correlation between low expression levels of HSD17B enzymes involved in E2 synthesis combined with high expression levels of E1-synthetizing HSD17B enzymes. As shown in Figure 8A–D, changes in the intratumoural expression levels of these enzymes, resulting in an increase in the E1:E2 ratio, are associated with a worse patient outcome for ER+ ovarian cancer disease.

## 3. Discussion

While obesity is a risk factor for many cancer types, its association with ER+ cancer development in women has been shown to be dependent on menopausal status, indicating a key role of estrogens in this process. It is known that the obese adipose tissue mediates inflammation, tumour growth and progression in ER+ cancer and that NF-κB pathway activation plays a key role in these processes [20,21]. It has been also proven in many studies that E2, the main estrogen in serum before menopause, inhibits the NF-κB pathway [24]; however, the role of E1, the major estrogen in the bloodstream after menopause and whose levels increase with obesity as it does the risk of developing ER+ cancers, has not been explored until date. This work aims to better understand the roles of E1 and E2 on NF-κB pathway regulation and tumourigenesis in ER+ cancers. 

For this purpose, we exposed the immortalized epithelial human cell line HeLa to E1 or E2 with or without the addition of TNF-α, a well-known NF-κB activator which is elevated in the obese adipose tissue [30]. We first checked the expression of ER and found that, even when the levels were quite low, ER was expressed in these cells. In addition, we performed additional experiments with HeLa cells transfected to overexpress the ERα (HeLa pERα) and demonstrated the successful transfection using the observed increased expression of GREB1, a direct target gene of ERα [31], after E2 treatment in HeLa pERα cells compared to the WT.

NF-κB plays an important role in obesity-mediated inflammation through the upregulation of pro-inflammatory cytokine expression [30,32]. As expected, NF-κB was activated and pro-inflammatory cytokines such as IL-6, IL-8 and CCL-2 increased in HeLa cells exposed to TNF-α. Interestingly, our results demonstrate that the addition of E1, but not E2, cooperates with TNF-α to mediate NF-κB activation and inflammation. Although E2 significantly decreases only TNF-α-driven IL-8 upregulation, these results are in concordance with a previous observation about the opposing effect of E2 in inflammation found in several diseases such as multiple sclerosis or arthritis [33]. In addition, E2 also represses IL-6 gene expression inhibiting NF-κB DNA binding to the IL-6 promoter [34], but we were not able to appreciate such repression in HeLa cells either with the 4 h treatment or with the treatment over time. Altogether, these results provide the first proof to date of a differential role of E1 and E2 in NF-κB regulation and obesity-mediated inflammation. 

IL-6, IL-8 and CCL-2 cytokines are mediators of tumour invasion and metastasis, and their increased levels are associated with cancer stem cell (CSC) enrichment and greater tumour stage and grade in breast cancer [34]. It is well known that NF-κB, which drives pro-inflammatory cytokine expression, also actively mediates tumourigenesis. Our results suggest that, in the context of obesity-associated inflammation, in vitro exposure of HeLa cells to TNF-α increased stemness markers. In this regard, we observed that TNF-α treatment increases ALDH activity in HeLa cells, markers commonly used to identify cervical CSC [35]. However, these results are not robust enough to confirm this behaviour. It is important to note that ALDH activity has also been associated with CSC features in different solid cancers, such as breast cancer [36]. Moreover, our results indicate that E1 and E2 have an opposite effect on TNF-α-driven ALDH activity upregulation. In particular, it seems that E1 maintains and E2 decreases ALDH enrichment mediated by TNF-α; however, additional experiments are needed to conclude this observation. In addition to ALDH activity, another widely used test to determine CSC enrichment is the upregulation of ES-TF expression, key transcription factors involved in normal and CSC self-renewal [37]. In this regard, it has been shown that their upregulation in differentiated cells is enough to give rise to the generation of induced pluripotent stem cells (iPS) [38]. Our results demonstrate that TNF-α + E1 treatment upregulated the expression of several ES-TFs in epithelial cells. In this sense, it has been previously shown that constitutive upregulation of c-MYC and KLF4, as seen in TNF-α + E1 exposed HeLa cells, is associated with CSC enrichment and tumour aggressiveness [39]. On the contrary, our results demonstrate that E2 downregulates c-MYC expression as was previously suggested by Maminta et al. [40]. In conclusion, our results indicate that E1 activates and E2 represses stemness feature enrichment in these cancer epithelial cells, particularly in the context of obesity-mediated inflammation. 

Another key important event in tumourigenesis and CSCs behaviour is the activation of the EMT program [41], a phenomenon in which NF-κB activation is also implicated [42]. For example, it has been shown in mammary epithelial cells that EMT is activated by NF-κB p65 subunit overexpression and TWIST1-driven upregulation, a key transcription factor involved in this process [43]. Similarly, Dong et al. observed that NF-κB/Twist1 overexpression by TNF-α treatment promotes HeLa migration and EMT activation [44]. Our results do not corroborate these previous observations with TNF-α exposure alone but show an upregulation of TWIST1 by TNF-α when combined with E1, but not E2, treatment. Moreover, we show that TNF-α + E1 treatment also upregulates the expression of other important EMT-driven transcription factors, such as SNAIL and SLUG, and confirmed the acquisition of a mesenchymal-like phenotype using the increased expression of the mesenchymal markers Vimentin and N-Cadherin [45]. Interestingly, our results also demonstrate that E1 treatment by itself is able to upregulate the expression of the key EMT markers SNAIL and TWIST1, which further support our findings about the key role of E1 in promoting the EMT process. In contrast, E2 treatment alone or in combination with TNF-α did not show an effect on EMT activation or on the acquisition of a mesenchymal phenotype in HeLa cells [46]. 

For decades, it has been assumed that E2 and E1 had similar biological functions, only differing in their ability to bind the ER, which was thought to be greater for E2. Thus, almost all works published in the field to date have investigated the role of E2, which is considered the active form of estrogens. Our work demonstrates that E2 and E1, the main estrogens in the serum of pre- and postmenopausal women, respectively, play different roles in obesity-mediated NF-κB activation and, therefore, inflammation, stemness and EMT regulation in ER+ epithelial cancer cells. In fact, here, we show that the key oncogenic role observed for E1 in ER+ breast cancer [25,26], which is also applicable to other ER+ cancers. 

These results are the first explanation to date of the differential association observed in the latest epidemiological studies between obesity and ER+ cancer risk in pre- and postmenopausal women. In particular, we have demonstrated that, contrary to the proven protective role of E2 in NF-κB activation and inflammation as seen in several diseases, in cancer, E1 cooperates with TNF-α to drive NF-κB activation, inflammation and tumourigenesis. These results have important clinical implications since they support the development of new hormone replacement therapies with reduced E1 levels for the potential reduction of ER+ cancer incidence in obese postmenopausal women. Moreover, these results also open a field of new drug development for ER+ cancers. Our work suggests that targeting E1 or its synthesis could be a potential strategy to design new therapies for ER+ cancer patients in the future.

Regarding the role of intratumoural E1 and E2 levels in ER+ cancer progression, results obtained from the analysis of the correlation between the expression of different HSD17B family members and ovarian cancer patient outcomes [47] indicate that increased intratumoural E1 levels are associated with a worse prognosis. Downregulation of E2 HSD17B-synthetizing enzymes or upregulation of E1 HSD17B-synthetizing enzymes were analysed using the KM Plotter^TM^ primary human ovarian cancer database to study their relationship with disease outcome. Of these, we analysed HSD17B1, HSD17B5, HSD17B7 and HSD17B12, known to have a reductive activity and to be involved in E2 synthesis, and HSD17B2, HSD17B4, HSD17B6, HSD17B10 and HSD17B14 with a preferential oxidative activity related to E1 synthesis [28,29]. Our results demonstrate a positive association between HSD17B5 expression with PFS, overexpression of HSD17B1 and HSD17B12 for PPS and overexpression of HSD17B7 for overall survival, all of which are reducing enzymes involved in E2 synthesis. This suggests a protective effect of the E2 biosynthetic pathway. Meanwhile, for the oxidizing enzymes involved in E1 biosynthesis, tumoural overexpression of HSD17B4, HSD17B6 and HSD17B14 resulted in lower ovarian cancer PFS; overexpression of HSD17B4 and HSD17B6 in ovarian primary tumours were related to decreased PPS, and high expression of the oxidative enzymes HSD17B4, HSD17B6 and HSD17B14 correlated with reduced OS. In this case, the E1 biosynthetic pathway was related to a worse outcome in these patients. 

Previous articles described a relationship between patient outcome and HSD17B enzyme expression for some ER+ cancers [25,26,28,48,49,50,51,52,53,54,55]. For example, high intratumoural expression of HSD17B enzymes involved in E1 synthesis, such as HSD17B2, HSD17B10 and HSD17B14, are associated with lower OS and/or DMFS in ER+ breast cancer patients [25,26]. On the contrary, increased expression of E2 HSD17B-synthetizing enzymes HSD17B1, HSD17B7 and HSD17B5 correlates with better ER+ breast cancer patient outcome for RFS and/or DMFS [26]. Regarding other ER+ cancer types, HSD17B2 expression and its different polymorphisms showed an inverse relationship with PFS and OS in prostate cancer [48] in patients with ERα-positive tumours. In adrenocortical carcinoma (ACC) the overexpression of HSD17B4 had an average ∼50 days longer OS than patients without [49]. Regarding HSD17B6, the OS and PFS increased with its expression in liver cancer [50], while for prostate cancer, it resulted in a worse prognosis. Negative regulation of E2-synthesizing enzymes or positive regulation of E1-synthesizing enzymes is also found in the results obtained from the TCGA program database for UCEC. Specifically, results were analysed for the enzymes HSD17B2, HSD17B10 and HSD17B12. The first two are responsible for the synthesis of E1 from E2 and the last one is responsible for the opposite process. In this sense, we found hypomethylation of the enzyme HSD17B2 in cancer samples, which, as expected, also showed a higher level of expression in samples of uterine tumour tissue. We also found that the enzyme HSD17B10 was overexpressed in tumour tissue compared to normal tissue. In contrast, the HSD17B12 enzyme showed higher methylation in normal tissue samples compared to tumour tissue. These results once again support the differential role of E1 and E2 in the regulation of ER+ breast cancer progression. In other studies, the overexpression of HSD17B7 was related to the progression of gastric cancer [51] and the poor prognosis of renal cancer [52]. Zhang et al. demonstrated that the expression of HSD17B11 promoted colorectal progression [53]. In contrast to our findings, Szajnik et al. suggested that HSD17B12 overexpression is related to a poor prognosis for ovarian carcinoma [54], but Etienne et al. found no relationship with PFS for breast cancer [55]. Meanwhile, for HSD17B14, higher expression was prognostic for poor survival in women with ER+ breast cancer [25]. These results indicate the importance of further analysis to demonstrate the main role of these isoforms in each cancer type.

## 4. Materials and Methods

### 4.1. Cell Lines and Cultures

HeLa cells were used as a cancer cell model (ATCC CCL2). HeLa is a human epithelial cell line derived from the cervical cancer cells of 31-year-old women. HeLa adherent cells were maintained at standard culture conditions in a humid incubator at 37 °C and 5% CO_2_ (Thermo Fisher Scientific, MO, USA), with Dulbecco’s Modified Eagle’s Medium (DMEM) (Sigma-Aldrich, St. Louis, MO, USA) supplemented with 10% heat-inactivated fetal bovine serum (FBS) (Thermo Fisher Scientific, MO, USA) and 1% stock solution of Penicillin/streptomycin (P/S), containing 10.000 U/mL Penicillin-G and 10 mg/mL streptomycin (Sigma-Aldrich, St. Louis, MO, USA) in 75 cm^2^ flask culture (Falcon). Media was renewed every 2 days and the subculturing was realized at 70–90% cell confluency.

For the different experiments, cells were treated with 10 ng/mL TNF-α, 10 nM E1, 10 nM E2, TNF-α + E1 and TNF-α + E2, as indicated for each assay. For control cells, the vehicle used for estrogen dilution, EtOH, was added. The experiments were carried out after 4 h or long exposure (1–3 weeks exposure) treatments. Before the treatments, cells were cultured in phenol red-free media (DMEM/F-12, HEPES, no phenol red, Thermo Fisher Scientific, Berkeley, MO, USA), since this compound has estrogenic activity and can compete with estrogens binding to ER [56], containing 5% charcoal-stripped fetal bovine serum (Gibco^TM^, Thermo Fisher Scientific, Berkeley, MO, USA) for at least 3 days. 

### 4.2. Protein Isolation and Western Blot

To study ERα expression, HeLa cells were washed with PBS and lysed for 15 min in RIPA buffer (Santa Cruz Biotechnology). Protein concentration was determined using a Pierce BCA Protein Assay kit (Thermo Scientific, Waltham, MA, USA). Proteins were loaded and run in 10% SDS-polyacrylamide gel. Proteins were transferred into PVCF membranes (Millipore Corp., Bedford, MA, USA) using electroblot transfer and blocked with 1X TBS-T + 5% BSA for 1 h at room temperature. Membranes were incubated with primary antibody anti-human ERα or anti-human GAPDH (Cell Signaling Technology) overnight at 4 °C. Antibodies were diluted 1:1000 in 1X TBS-T + 5% BSA. The primary antibody was then washed three times, and membranes were incubated with the secondary antibody anti-rabbit IgG, diluted 1:10.000. Protein–antibody complexes were visualized using chemiluminescence (Pierce ECL, Thermo Scientific) with the program IMAGE READER LAS-4000 in a LAS-4000 imaging system.

### 4.3. RNA Isolation and Real-Time PCR Analysis

Total cellular RNA was isolated using Trizol (TRI Reagent) according to the manufacturer’s instruction (Sigma-Aldrich). Total RNA (1 μg) was reverse transcribed using the Reverse Transcription System (Promega) following the manufacturer’s protocol. The resulting cDNA was diluted to a final volume of 100 μL with nuclease-free water and 2.5 μL were used for each subsequent real-time PCR reaction. Quantitative PCR was performed using the GoTaq qPCR Master Mix, SYBR Green (Promega) on a 7500 device for different target genes (ESR1, ESR2, GREB1, CCL2, IL-6, IL-8, SLUG, SNAIL, TWIST1, Vimentin, N-Cadherin, c-MYC, SOX2 and KLF4) and the housekeeping gene GAPDH. Sequences of the primers used are shown in Table 1. Triplicates of each reaction were carried out. The expression levels were normalized with the corresponding GAPDH values and are shown as fold changes relative to the control sample. 

### 4.4. Flow Cytometry

Aldehyde dehydrogenase (ALDH) enzyme activity was assayed with an AldefluorTM kit assay (Stemcell Technologies, Vancouver, BC, Canada) following the manufacturer’s instructions. Labelled cells were acquired using a FACS CANTO II cytometer (BD Biosciences) for analysis. 

For CD133 analysis, cells were detached with trypsin, washed with PBS, resuspended in a blocking buffer solution (PBS supplemented with 2% bovine serum albumin (BSA)) and then stained with anti-human CD133-APC antibody or IgG control (Miltenyi Biotec, Auburn, CA, EEUU). After 15 min incubation at 4 °C, the dark cells were washed and acquired using a FACS CANTO II cytometer (BD Biosciences) for analysis.

### 4.5. MTT Assay

Cells were seeded into 96-well culture plates at a density of 2000 cells/well and grown in DMEM + 10% FBS + 1% P/S for 24 h before neomycin addition. Different concentrations of neomycin (0 to 80 μg/mL and 200 to 600 μg/mL) were added and an MTT assay was performed after 48 h following manufacturer´s instructions. Cell viability was determined by measuring absorbance at 570 nm using a plate reader (Promega).

### 4.6. Immunofluorescence

Cells were plated in 96-well culture plate and after 45 min of TNF-α treatment were fixed in 4% paraformaldehyde. Cells were washed, incubated with 0.1% PBS-Triton, blocked with 0.5% BSA in PBS and incubated overnight at 4 °C with a 1:150 dilution of the primary antibodies against human ERα, RelA, RelB, cRel, p100/p52, p105/p50 (Cell Signaling Technology, Europe, B.V). Then, cells were washed 3 times and incubated with the secondary antibody FITC-conjugated anti-rabbit diluted 1:200. Nucleus cells were stained with a DAPI (4′,6-diamidino-2-phenylindole) solution and pictures were taken with fluorescent microscopy (Leica). As an internal control, cells were incubated with a secondary antibody without prior incubation with the primary antibody. 

### 4.7. Cells Transfection

HeLa cells were seeded in a 6-well plate and transfected with an ERα WT cDNA plasmid (GeneCopoeia) for the constitutive overexpression of ERα. Cell transfection was realized using Lipofectamine 3000^TM^ and following the manufacturer´s instructions. Transfected cells were selected by 800 μg/mL neomycin treatment.

### 4.8. Statistical Analysis

All graphed data are presented as mean ± SEM from at least three biological replicate experiments. The Student’s *t*-test was used for experiments with two groups. Comparisons of >2 groups used one-way analysis of variance (ANOVA) followed by Dunnett´s or Tukey´s post hoc analysis. Some experiments used a two-way analysis of variance followed by Tukey´s post hoc tests.

### 4.9. KM-Plotter and UALCAN Analysis

We carried out a bioinformatic analysis of data from ovarian cancer patients for the association of the expression levels of different HSD17B enzymes and progression-free survival (PFS), post-progression survival (PPS) and overall survival (OS) using the KM Plotter^TM^ primary human ovarian cancer database [57]. Briefly, cases are separated into two groups based on tumoural mRNA expression levels of a certain gene, including the high and low expression groups. Survival curves are produced according to the life table method described by Kaplan and Meier. Differences are estimated with the log-rank test, which calculates the chi-square (*X^2^*) for each event time for each group and sums the results. In addition, hazard ratios are provided, which give a relative event rate comparing both groups. 

In addition, we use the UALCAN database to obtain the UCEC data. UALCAN is an interactive web resource for analysing cancer OMICS data. UALCAN provides an expression analysis option using data from The Cancer Genome Atlas (TCGA) dataset. We evaluated the expression of different genes from the family HSD17B [58].

## 5. Conclusions

Our work is the first to show that E2 and E1, the main estrogens in the serum of pre- and postmenopausal women, respectively, play different roles in obesity-mediated NF-κB activation and, therefore, inflammation, stemness and EMT regulation in ER+ epithelial cells. These results are the first explanation to date of the differential association observed in the latest epidemiological studies between obesity and ER+ cancer risk in pre- and postmenopausal women. In particular, we have demonstrated that, contrary to the proven protective role of E2 in NF-κB activation and inflammation as seen in several diseases, E1 cooperates with TNF-α to drive NF-κB activation, inflammation and tumourigenesis. These results have important clinical implications since they support the development of new hormone replacement therapies with reduced E1 levels for the potential reduction of ER+ cancer incidence in obese postmenopausal women. Moreover, these results also open a field of new drug development for ER+ cancers. In this regard, current clinical strategies for ER+ breast cancer treatment are based in the modulation or downregulation of ER and in the inhibition of aromatase, the main enzyme involved in estrogen synthesis. Our work indicates that targeting E1 or its synthesis could be a potential strategy to design new therapies for ER+ cancer patients in the future.

## Figures and Tables

**Figure 1 ijms-24-01221-f001:**
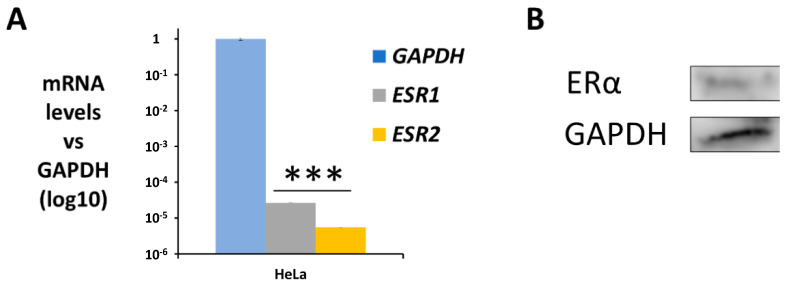
ER expression in HeLa cells. (**A**) Gene expression profile of both ER isoforms using qPCR, graphed relative to the housekeeping GAPDH as mean ± SEM (*n* = 3). (**B**) Western Blot of ERα expression using GAPDH as housekeeping (*** *p* < 0.001).

**Figure 2 ijms-24-01221-f002:**
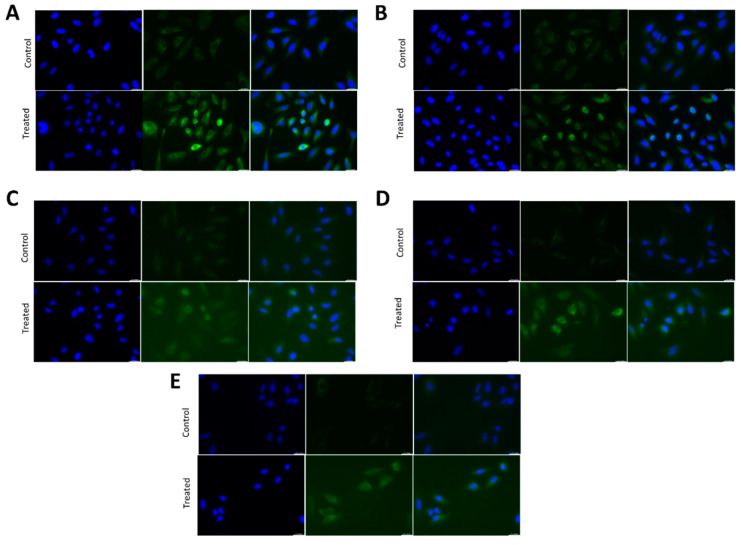
NF-κB nuclear translocation using TNFα treatment in HeLa cells. (**A–E**) Fluorescent immunostaining of HeLa cells untreated (control) or treated with 10 ng/mL TNF-α for 45 min using antibodies against human RelA/p65 (**A**), RelB (**B**), cRel (**C**), p50 (**D**), p52 (**E**) (green). Nuclei were stained with DAPI (blue).

**Figure 3 ijms-24-01221-f003:**
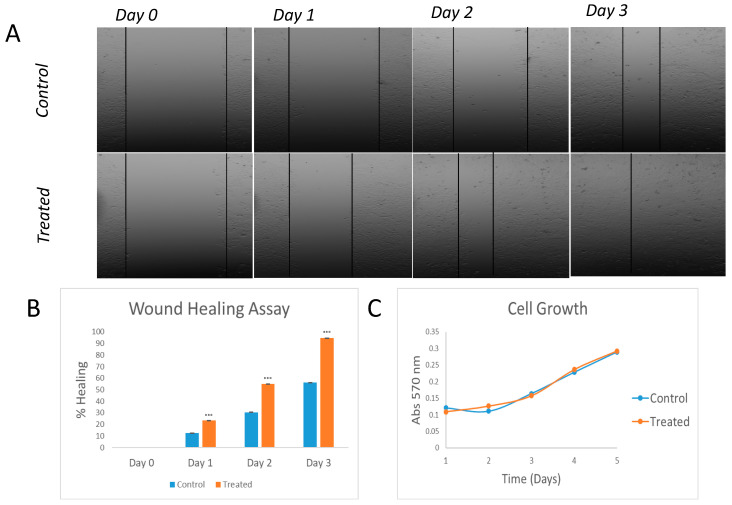
Wound-healing and proliferation assays in TNFα-treated in HeLa cells. Images of wound-healing assay performed on HeLa cells. Treated vs control (**A**) and graph representation (*n* = 3; *** *p* < 0.001 vs control) (**B**). Study of cell growth differences between treated and untreated cells (**C**).

**Figure 4 ijms-24-01221-f004:**
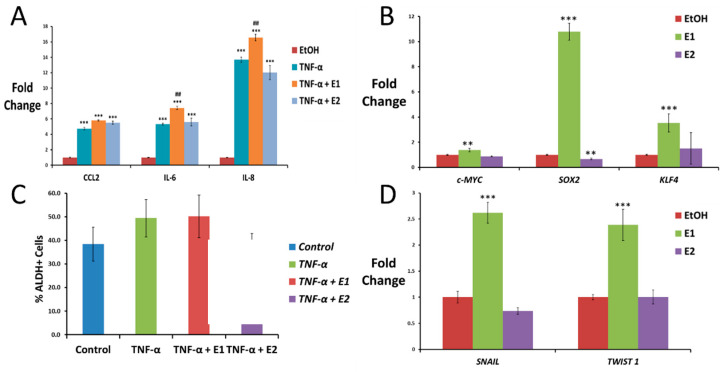
Effect of TNF-α, estrone and estradiol on driving inflammation, stemness and EMT in HeLa cells. (**A**) Pro-inflammatory cytokine expression measured using qPCR in HeLa cells after treatment with 10 ng/mL TNF-α for 4 h alone or in combination with 10 nM E1 or E2 (data normalized to 1 for control (EtOH) using GAPDH as the housekeeping gene). (**B**) qPCR for the expression of the ES-TF c-MYC and SOX2 in HeLa cells after 3 weeks of exposure to 10 nM E1 or E2 (data normalized to 1 for control (EtOH) using GAPDH as the housekeeping gene). (**C**) Expression of ALDH activity measured using flow cytometry in HeLa cells untreated (control) or treated with 10 ng/mL TNF-α alone or in combination with 10 nM E1 or E2 for 3 weeks. (**D**) EMT transcription factor expression measured using qPCR in HeLa cells exposed to 10 nM E1 or E2 for 3 weeks (data normalized to 1 for control using GAPDH as the housekeeping gene). All data are graphed as mean ± SEM from experiments performed in triplicates and repeated at least 3 times. ** *p* < 0.01 *** *p* < 0.001 vs control; ## *p* < 0.05 vs *TNF-α*.

**Figure 5 ijms-24-01221-f005:**
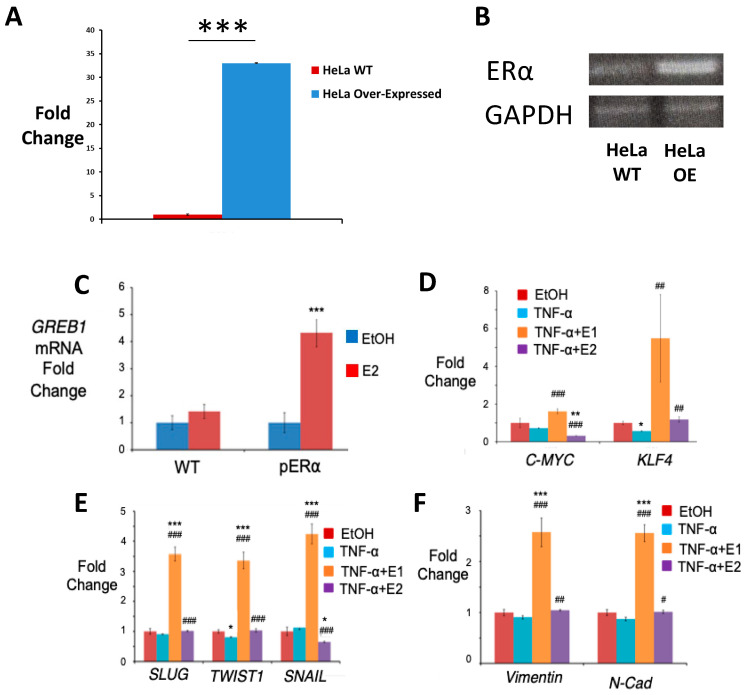
Effect of TNF-α, estrone and estradiol on driving stemness and EMT in HeLa pERα cells. (**A**) Western Blot of ERα expression using GAPDH as housekeeping. (**B**) Gene expression profile of ERα using qPCR, graphed as mean ± SEM (*n* = 3) using WT as normalizer and GAPDH as housekeeping gene (*** *p* < 0.001 vs Wild Type). (**C**) qPCR for the expression of GREB1 in WT and pERα HeLa cells treated with 10 nM E2 for 4 h (data normalized to 1 for control (EtOH) using GAPDH as the housekeeping gene). (**D**) ES-TF expression in HeLa pERα cells assayed using qPCR after exposure to 10 ng/mL TNF-α alone or in combination with 10 nM E1 or E2 for at least 1 week (data normalized to 1 for control using GAPDH as the housekeeping gene). (**C**,**D**) qPCR analysis for the expression of the EMT transcription factors SLUG, TWIST1 and SNAIL (**E**) and the EMT markers Vimentin and N-Cadherin (**F**) in HeLa pERα cells treated with the vehicle (EtOH, control) or 10 ng/mL TNF-α alone or in combination with 10 nM E1 or E2 for 3 weeks (data normalized to 1 for control using GAPDH as the housekeeping gene). All data are graphed as mean ± SEM from experiments performed in triplicates and repeated at least 3 times. * *p* < 0.05 ** *p* < 0.01 *** *p* < 0.001 vs control; # *p* < 0.05 ## *p* < 0.01 ### *p* < 0.001 vs TNF-α.

**Figure 6 ijms-24-01221-f006:**
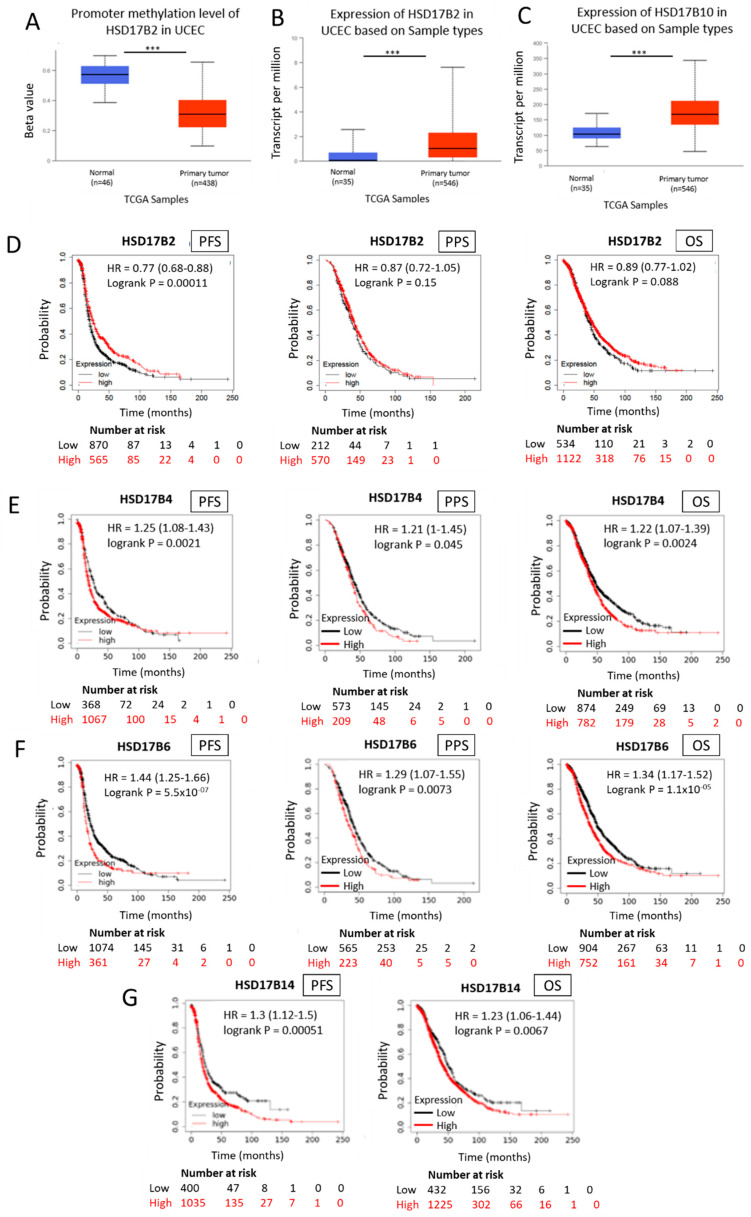
Expression and methylation of HSD17B enzymes involved in estrone synthesis and ovarian cancer patient outcome (**B**–**F**). Methylation levels of the HSD17B2 enzyme in normal and tumour tissue samples obtained from “The Cancer Genome Atlas” (TCGA) database (**A**). Expression levels of the HSD17B2 enzyme in normal and tumour tissue samples obtained from the TCGA (**B**). Expression levels of the HSD17B10 enzyme in normal and tumour tissue samples obtained from the TCGA (**C**). Kaplan–Meier curves comparing low (black line) vs high (red line) tumoural expression levels of HSD17B2, HSDD17B4 and HSD17B6 enzymes involved in E1 synthesis and progression-free survival (PFS), post-progression-free survival (PPS) and overall survival (OS) in ovarian cancer patients (HR = Hazard Ratio) (**D**–**F**) in the case of HSD17B14 only PFS and OS are represented (**G**). *** *p* < 0.001.

**Figure 7 ijms-24-01221-f007:**
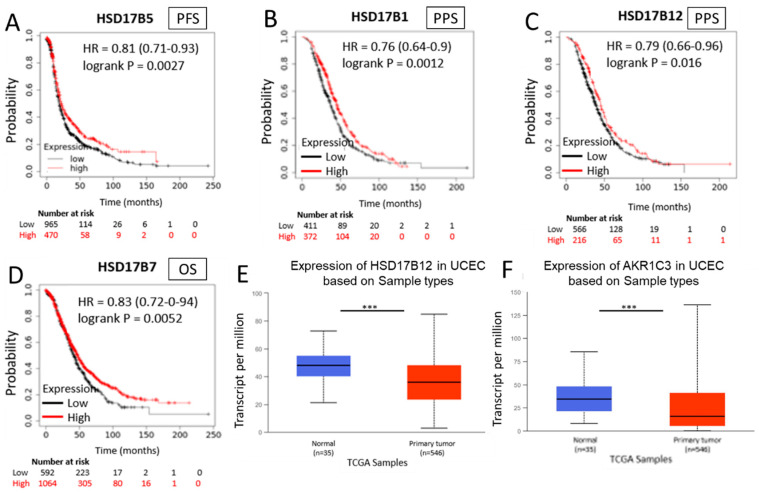
Expression of HSD17B enzymes involved in estradiol synthesis and ovarian cancer patient outcome (**A**–**D**)**.** Kaplan–Meier curves comparing low (black line) vs high (red line) tumour expression levels of different enzymes of the HSD17B family; HSD17B5 enzyme in PFS (**A**); HSD17B1 enzyme in PPS (**B**); HSD17B12 enzyme in PPS (**C**); HSD17B7 enzyme in OS (**D**). Expression of HSD17B12 in normal and tumour samples taken from the TCGA program (**E**). Expression of AKR1C3 in normal and tumour samples taken from the TCGA program (**F**). *** *p* < 0.001.

**Figure 8 ijms-24-01221-f008:**
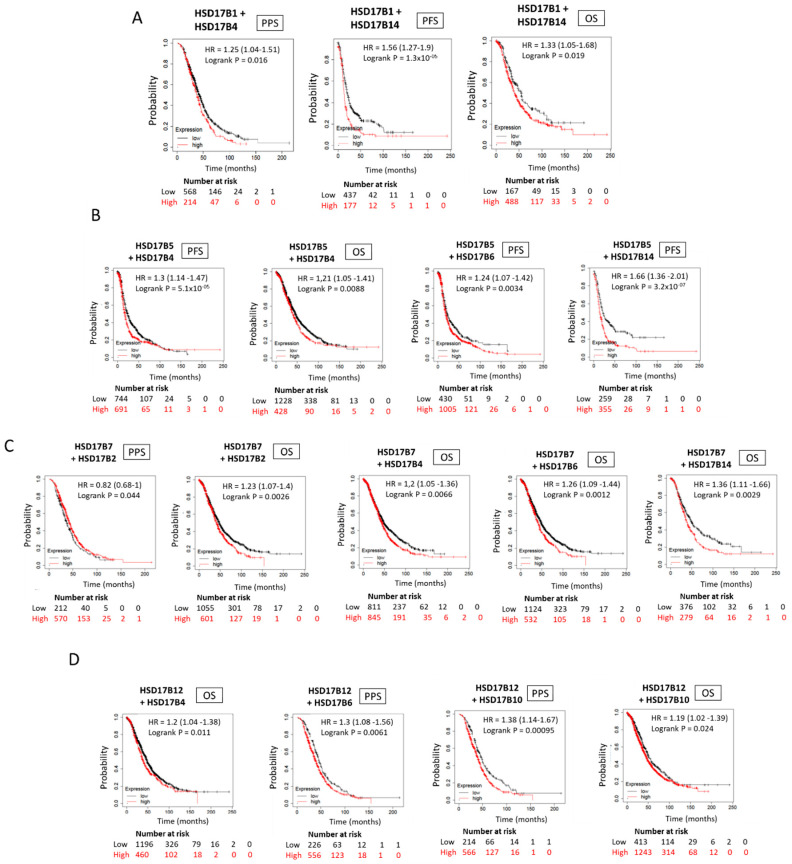
Ovarian cancer patient outcome based on combined low and high expression of E2- and E1-synthetizing HSD17B enzymes, respectively. (**A**–**D**) Kaplan–Meier curves comparing the correlation between low intratumoural expression levels of the E2-synthetizing enzymes HSD17B1 (**A**), HSD17B5 (**B**), HSD17B7 (**C**) and HSD17B12 (**D**); combined with high intratumorual expression of the different E1-synthetizing HSD17B isoforms indicated, including HSD17B2, HSD17B4, HSD17B6, HSD17B10 and HSD17B14; and PPS, PFS and OS in ovarian cancer patients.

**Table 1 ijms-24-01221-t001:** Sequence of primers used in PCR studies.

Gene	Primers Secuence
ESR1	Fw: 5′- GCTTACTGACCAACCTGGCAGA -3′ Rev: 5′- GGATCTAGCCAGGCACATTC -3′
ESR2	Fw: 5′- ATGGAGTCTGGTCGTGTGAAGG -3′ Rev: 5′- TAACACTTCCGAAGTCGGCAGG -3′
GREB1	Fw: 5′- CAATTCCATCGAGGCATCC -3′ Rev: 5′- GGCTACCACCTTCTAGAGC -3′
CCL2	Fw: 5′- AAGAAGCTGTGATCTTCAAGAC -3′ Rev: 5′- CCATGGAATCCTGAACCCA -3′
IL-6	Fw: 5′- GATTCAATGAGGAGACTTGCC -3′ Rev: 5′- TGTTCTGGAGGTACTCTAGGT -3′
IL-8	Fw: 5′- TGCCAAGGAGTGCTAAAG -3′ Rev: 5′- CTCCACAACCCTCTGCAC -3′
SNAIL	Fw: 5′- TGCCCTCAAGATGCACATCCGA -3′ Rev: 5′- GGGACAGGAGAAGGGCTTCTC -3′
SLUG	Fw: 5′- ATCTGCGGCAAGGCGTTTTCCA -3′ Rev: 5′- GAGCCCTCAGATTTGACCTGTC -3′
TWIST1	Fw: 5′- GCCAGGTACATCGACTTCCTCT -3′ Rev: 5′- TCCATCCTCCAGACCGAGAAGG -3′
N-Cad	Fw: 5′- CCTCCAGAGTTTACTGCCATGAC -3′ Rev: 5′- GTAGGATCTCCGCCACTGATTC -3′
VIMENTIN	Fw: 5′- AGGCAAAGCAGGAGTCCACTGA -3′ Rev: 5′- ATCTGGCGTTCCAGGGACTCAT -3′
c-MYC	Fw: 5′- CCTGGTGCTCCATGAGGAGAC -3′ Rev: 5′- CAGACTCTGACCTTTTGAACGG -3′
KLF4	Fw: 5′- CATCTCAAGGCACACCTGCGAA -3′ Rev: 5′- TCGGTCGCATTTTTGGCACTGG -3′
SOX2	Fw: 5′- GCTACAGCATGATGCAGGACCA -3′ Rev: 5′- TCTGAGAGCTGGTCATGGAGTT -3′
GAPDH	Fw: 5′- ATCAAGTGGGGCGATGCTG -3′ Rev: 5′- ACCCATGACGAACATGGGG -3′

## Data Availability

Not applicable.

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
