# Peer review of "Estradiol and Estrone Have Different Biological Functions to Induce NF-κB-Driven Inflammation, EMT and Stemness in ER+ Cancer Cells"

_ijms, 2023, doi:10.3390/ijms24021221_

Round 1

Reviewer 1 Report

In this study, Diaz et al. revealed that estradiol (E2) and estrone (E1) have different biological functions in inflammation-induced ER+ cancer cells. The study clearly shows that E1 increases embryonic stem-transcription factors expression and induces epithelial-to-mesenchymal transition in estrogen receptor positive HeLa cells. The authors demonstrated that overexpression of E2 synthetizing HSD17B isoforms are associated with better outcome, for patients diagnosed with ER+ ovarian and uterine corpus carcinomas. Although the study is original and meaningful, the design is reasonable, and the methods are appropriate, several points need to be clarified and these must be corrected as following;

11. Figure 1C: The low fluorescence levels is quite surprising. The ERa fluorescence light level should be improved and, if possible, presented with the "Merge" method. The “GAPDH” on the DAPI staining picture should be removed and replaced with "DAPI".

22. Figure 2: The protein levels of NFkB transcription factors were shown after of TNF-a treatment for 45 min. Is short-term TNF-a treatment enough for the expression of a protein? Could the insufficiently low level of fluorescence be related to the TNF-a treatment time? Fig.2 should be better presented.

33. Figure3C: The location of labeling should be shown either in x-axis or at the side of diagram as legend.

44. Line 181:” As shown in Figure 3E, long exposure to E1…” I guess the Figure3E should be rewritten as Figure 3D.

55. Figure 5: The promoter methylation level of HSD17B2 should be written in figure legend.

66. Line 307-308: “It has been also proven in many studies that E2, the main estrogen in serum before menopause, inhibits…” The studies should be cited.

77. Line 320-321: the role of NF-κB in obesity-mediated inflammation should be cited.

88. The focus of this study is the different effects of E1 and E2 on cancer cells, however, proliferation and migration, which are important parameters in cancer studies, have not been performed. Is there any reason for this?

99. Line 446: Is the concentration of TNF-a 10µg/ml or 10ng/ml?

110. Line 458: the right name of membrane is “polyvinylidene difluoride (PVDF) membranes”

111. In mehod section: It is mentioned in the method section that MTT assay was performed using different conc. of neomycin. However, it is not clear why the study was performed, and there is no data in the results section. This should be improved.

Reviewer 2 Report

The manuscript by Diaz-Ruano et al. focused on the potential effects of estrone (E1) and estradiol (E2) on inflammation, EMT, and stemness if various ER+ cancers. The work is of some interest although I feel the lack of some mechanistic outputs (proliferation, migration) to be of considerable concern and hinders the conclusions that can be made about the work. There are also necessary controls that have not been included. Thus, my major concerns on this paper are:

1)      For the HeLa pERalpha the authors do not show any evidence that ERalpha expression has indeed been increased. A  western blot and qRT-PCR for ERalpha in these cells should be presented to show this.

2)      The dose of TNFalpha used in the HeLa studies is extremely high at 10ug/mL. What is the rationale for using such a high dose which is far outside the physiological range? What would happen to some of these inflammatory or EMT markers if TNFalpha was at a more physiological dose?

3)      Although it is interesting that the combination of TNFalpha and E1 causes increased changes in inflammatory and EMT markers, the authors do not show any assays demonstrating a mechanistic outcome from these studies. The authors mention metastasis as a potential outcome but show no evidence for this. I think this needs to be address and a functional output should be explored. Does TNFalpha plus E1 lead to greater metastasis as measured by a scratch assay? Or is there increased proliferation?

4)      The authors do not show the expression of HSD17B enzymes in the HeLa cells. What is happening the E1 or E2 metabolism when these estrogens are placed on the cells? The results with regards inflammatory and EMT markers may not be as obvious as there may be conversion of E1 to E2 (or vice versa). Can this be checked?

5)      The authors examine the HSD17B family of enzymes in both TCGA and KM-plotter in various cancers, looking at overall progression-free survival, post-progression free survival, and overall survival. This is a rather crude way of looking at whether these enzymes are important in these patient outcomes. As this family of enzymes does both reactions, taking individual enzymes and plotting them over-simplifies the actual estrogen metabolism that may be happening. Is there a way to look at how HSD17B expression effects patient outcomes when the reverse reaction enzymes expression is low? For example, could you look at HSD17B1 expression in only patients that have mid-to-low HSD17B2, HSD17B4 and HSD17B6 expression? I understand this will limit n number, but if possible it may provide a better understanding of estrogenic effects.

6)      For Figure 5, why is the ovarian cancer outcomes for HSD17B2 not presented?
